# Preparation of TiO_2_ and Fe-TiO_2_ with an Impinging Stream-Rotating Packed Bed by the Precipitation Method for the Photodegradation of Gaseous Toluene

**DOI:** 10.3390/nano9081173

**Published:** 2019-08-16

**Authors:** Guangping Zeng, Qiaoling Zhang, Youzhi Liu, Shaochuang Zhang, Jing Guo

**Affiliations:** Shanxi Province Key Laboratory of Higee-Oriented Chemical Engineering, North University of China, Taiyuan 030051, China

**Keywords:** high-gravity technique, impinging stream-rotating packed bed, photocatalysis, precipitation method, nano titanium dioxide, doping, photocatalytic activity

## Abstract

Nano-TiO_2_ has always been one of the most important topics in the research of photocatalysts due to its special activity and stability. However, it has always been difficult to obtain nano-TiO_2_ with high dispersion, a small particle size and high photocatalytic activity. In this paper, nano-TiO_2_ powder was prepared by combining the high-gravity technique and direct precipitation method in an impinging stream-rotating packed bed (IS-RPB) reactor followed by Fe^3+^ in-situ doping. TiOSO_4_ and NH_3_·H_2_O solutions were cut into very small liquid microelements by high-speed rotating packing, and the mass transfer and microscopic mixing of the nucleation and growth processes of nano-TiO_2_ were strengthened in IS-RPB, which was beneficial to the continuous production of high quality nano-TiO_2_. Pure TiO_2_ and iron-doped nano-TiO_2_ (Fe-TiO_2_) were obtained in IS-RPB and were investigated by means of X-ray diffraction (XRD), Raman, scanning electron microscopy (SEM), transmission electron microscopy (TEM), X-ray photoelectron spectroscopy (XPS), ultraviolet-visible diffuse reflectance spectroscopy (UV-vis DRS) and Brunauer–Emmett–Teller (BET) analysis, which found that pure TiO_2_ had a particle size of about 12.5 nm, good dispersibility and a complete anatase crystal at the rotating speed of packing of 800 rpm and calcination temperature of 500 °C. The addition of Fe^3+^ did not change the crystalline structure of TiO_2_. Iron was highly dispersed in TiO_2_ without the detection of aggregates and was found to exist in a positive trivalent form by XPS. With the increase of iron doping, the photoresponse range of TiO_2_ to visible light was broadened from 3.06 eV to 2.26 eV. The degradation efficiency of gaseous toluene by Fe-TiO_2_ under ultraviolet light was higher than that of pure TiO_2_ and commercial P25 due to Fe^3+^ effectively suppressing the recombination of TiO_2_ electrons and holes; the highest efficiency produced by 1.0% Fe-TiO_2_ was 95.7%.

## 1. Introduction

Because of its high chemical stability and high photocatalytic activity, nano-TiO_2_ has great development prospects in the fields of environmental protection [1], such as in photocatalytic disinfection, antibacterial use [2,3], wastewater treatment [4], air purification [5], and so on. Especially in the face of the current difficulty of eliminating volatile organic compound (VOC) pollution in the air, TiO_2_ photocatalysis technology has shown unique advantages [6,7]. However, the preparation cost of nano-TiO_2_ is high, and the process is difficult to control, meaning that nano-TiO_2_ has poor dispersion and a wide particle-size distribution. Moreover, the photocatalytic performance restricts its development and application: firstly, the band gap of 3.2 eV limits it to using only ultraviolet light below 387.5 nm in wavelength [8]. Secondly, the electron and hole produced by the light excitation of TiO_2_ recombine very easily [9,10]. Therefore, developing preparation technology of TiO_2_ with good dispersion and a uniform particle size and exploring a way to broaden the light response range and inhibit the electron and hole recombination of TiO_2_ are the key problems which need to be solved urgently in order to develop the commercialization and advanced application of nano-TiO_2_.

The most common synthesis method of nano-TiO_2_ is the sol-gel method [11,12]. However, the raw material titanium alkoxide is relatively expensive, and the whole process not only produces a large number of organic compounds but also needs to use a large number of organic solvents, resulting in organic waste pollution, causing problems for the mass production of nano-TiO_2_. The preparation of nano-TiO_2_ by the precipitation method has the advantages of a cheap raw material, a simple process and a non-toxic organic solvent, which is convenient for industrial production. However, the traditional precipitation method generally uses an intermittent stirred tank with low mass transfer and microscopic mixing performance as the reactor. The nucleation and growth process occur before the homogeneous mixing of the titanium precursor and precipitant in the intermittent stirred tank, and there is a high level of non-uniformity in this process. It is difficult to move the completely grown particles from the intermittent stirred tank in time, resulting in those particles possibly being used as a crystal nucleus to grow further. As a result, the particle sizes are uneven and seriously agglomerate. High-gravity technology has been proven to be an excellent technology for the preparation of high-quality nanometer powder. The rotating packed bed (RPB) [13] is the typical reactor of high-gravity technology, which uses a motor to drive the packing to rotate at a high speed, resulting in a huge centrifugal force field. In the RPB, the raw material liquid is immediately cut and dispersed by the packing to form a very small liquid microelement, and a uniform microscopic mixing effect is obtained [14]. The nucleation and growth of the particles occur in a uniform supersaturation environment, and so the particles sizes are similar. The particles in a strong centrifugal force field will be immediately thrown out of the packing and separated from the raw materials and will not grow further, meaning that the particles are able to maintain good dispersion. A large number of studies have confirmed that RPB is an excellent reactor for the preparation of nanoparticles by the precipitation method, such as ZrO_2_ [15], Fe_3_O_4_ [16,17], Mg(OH)_2_ [18], and some composite materials [19,20], etc. However, there are few reports into the preparation of nano-TiO_2_ by RPB.

In order to improve the photocatalytic performance of TiO_2_, metal doping has always been favored by the research community [21,22]. Previous studies have shown that doping Mn, Fe, Cu, Co, Ag, Ce and other metal ions can accelerate the migration rate of photogenerated electrons and inhibit the recombination of electrons and holes to a certain extent [23]. In addition, it has been reported that metal ion doping contributes to the expansion of the optical absorption range of TiO_2_ [24]. U Khalid et al. [25] reported that the band gap of TiO_2_ decreased to 2.67 eV after doping Mn^2+^.

In this paper, TiO_2_ and Fe^3+^ ions with in-situ doped TiO_2_ (Fe-TiO_2_) nanoparticles were prepared by the direct precipitation method with an impinging stream-rotating packed bed (IS-RPB) [20,26,27] as a reactor and used for the photodegradation of gaseous toluene as a typical pollutant in air. The main purpose of this work is to make use of the efficient mass transfer capacity and strong microscopic mixing characteristics of IS-RPB to obtain nano-TiO_2_ with high dispersion and small particle size and to prepare Fe-TiO_2_ by uniformly doping Fe^3+^ in TiO_2_. The doping of Fe^3+^ shows the ability to enhance the response of TiO_2_ to visible light and reduce the rebinding of electrons and holes, which is beneficial to improving the photocatalytic performance. The morphology, structure and optical response of TiO_2_ and Fe-TiO_2_ were analyzed by various characterization methods. The photocatalytic activity of the photocatalyst was studied by the degradation of gaseous toluene as the typical VOC. Toluene was taken as the target VOC mainly because it is one of the most common pollutants both indoors or outdoors [28,29]. Furthermore, there have been many studies on the photocatalytic degradation of toluene, and thus there is a sufficient theoretical basis and practical feasibility [30,31,32].

## 2. Materials and Method

### 2.1. Materials

Titanyl sulfate (TiOSO_4_, 93%) was purchased from Shanghai Macklin Biochemical Co., Ltd. (Shanghai, China). Ammonium hydroxide solution (NH_3_·H_2_O, 25%–28%), ethanol anhydrous (99.5%), ferric chloride hexahydrate (FeCl_3_·6H_2_O, 99.0%) and other chemicals were obtained from Tianjin Guangfu Fine Chemical Research Institute (Tianjin, China). All reagents were used without any additional purification, and all stock and working solutions were prepared in deionized water.

### 2.2. Synthesis of Nano-TiO_2_

In order to prepare nano-TiO_2_, an IS-RPB reactor was used for the immediate precipitation process with NH_3_·H_2_O as a precipitant, as in the studies of Fan et al. [16] and Jiao et al. [33]. The details of the reactor are as follows: the packing was made by stainless steel wire meshes with a 2.5 mm diameter—the inter diameter, outer diameter and axial height of which, respectively, were 60 mm, 160 mm and 60 mm—and was connected to the motor by shaft. Two jet nozzles with a 1.5 mm diameter were employed in the opposite direction to cause solutions to be ejected to form an impinging stream. Others elements included the liquid inlet/outlet, a shell with a 200 mm diameter, et al.

The complete process was as follows: TiOSO_4_ at 0.2 mol/L and pH = 1 was prepared as solution A. NH_3_·H_2_O of 1.0 mol/L was prepared as solution B. Then, solution A and B separately were added to liquid storage tanks 1 and 2, and the solution entered the liquid distributor through the action of the pump; the flow rates were adjusted to be the same by a liquid flowmeter. After that, two stream flows were ejected at high speed from a jet nozzle in the opposite direction, producing an extended radial impinging spray surface, completing the first micromixing. Then, the impinging spray surface entered from the inner surface of the packing; under the action of centrifugal force, the liquid passed through the packing and was cut and dispersed by the high-speed rotating packing many times simultaneously to complete the second micromixing and precipitation reaction. Finally, the precipitates were collected and then separated using a high-speed centrifuge and were washed with deionized water and ethanol anhydrous to neutral pH, and no sulfate ion was detected by Ba^2+^. The precipitates were dried to obtain amorphous nano-TiO_2_. Then, nano-TiO_2_ was calcined at 400 °C to 800 °C to obtain a crystalline structure. The experimental setup is shown in Figure 1.

Previous studies [15,16] have shown that the rotating speed of packing has a great influence on the synthesis of nanomaterials by RPB or IS-RPB. Rotating packing is not only the source of a high-gravity field but also the most important and basic technical support of high-gravity technology. In this experiment, by changing the rotating speed of the packing to 400, 800 and 1200 rpm (according to the calculation formula of Fan et al. [16] and Jiao et al. [33], the corresponding high-gravity levels are 10.44, 41.78, 94.01), we investigated the effect of the rotating speed of packing in IS-RPB on the synthesis of nano-TiO_2_.

### 2.3. Synthesis of Fe-TiO_2_ Nanoparticles

Fe-TiO_2_ were prepared by adding a varying amount of FeCl_3_·6H_2_O into solution A, as described in Section 2.2, and other steps were repeated as per Section 2.2. In this experimental, 0.1351 g to 1.3510 g of FeCl_3_·6H_2_O were added to solution A; the molar ratios of Fe^3+^ to Ti^4+^ were 0.5%, 1.0%, 3%, 5%. The resulting Fe-TiO_2_ nanoparticles were called 0.5%Fe-TiO_2_, 1.0%Fe-TiO_2_ et al.

### 2.4. Characterization

The crystalline structures of the synthesized nanoparticles were detected using an X-ray diffractometer (XRD, DX-2700, China). The Raman spectra were determined using MultiRAM Raman spectroscopy (Bruker Corporation, Raman, Italy). The chemical states of the samples were determined using X-ray photoelectron spectroscopy (XPS, Escalab 250Xi, USA). The morphology, microstructure and particle size of the synthesized nanoparticles were measured using an SEM (HITACHI S4800, Tokyo, Japan) and a transmission electron microscope and high-resolution transmission electron microscopy (TEM and HRTEM, FEI Titan G2 60-300, USA) equipped with energy dispersive X-ray (EDX) spectroscopy. The optical absorption ranges of the samples were ascertained using UV-vis diffused reflectance spectroscopy (UV-vis DRS, Agilent cary 5000, USA).

### 2.5. Photocatalytic Activity

The photocatalytic activity was evaluated in a self-made quartz reactor (Figure 2; the volume is 3.2 L) at normal temperature and pressure. The environment in the reactor was consistent with indoor conditions. In the experiment, the photocatalyst was sprayed onto a glass plate by ethanol suspension, and then the glass plate was put into the quartz reactor. After the quartz reactor was sealed, 1.5 µL of liquid toluene (C_7_H_8_, 99%) was injected using a microsyringe (the toluene concentration was 105 ppm). The conversion formula is as follows:
(1)$$V_{T}=\frac{\frac{M}{V_{m}}{\times (C}_{V}\times 10^{-6})\times V_{R}}{\rho \times 99\%}$$
where V_T_ (µL) is the volume of liquid toluene, M (92.14 g/mol) is the molar mass of toluene, V_m_ (24.5 L/mol) is the molar volume of gas at normal temperature and pressure, C_V_ (ppm) is the gaseous toluene concentration, V_R_ (3.2 L) is the volume of the self-made quartz reactor, ρ (0.865 g/cm^3^) is the density of toluene, and 99% is the purity of liquid toluene. We waited for toluene to be mixed evenly, and a UV lamp (18 W, 254 nm) was turned on at the front of the irradiation reactor. The light intensity was 10.96 mW/cm^2^, measured by a PM120VA spectro-irradiator. The toluene concentration was detected by an online FID detector (SIGNAL GROUP, USA) at an interval of 30 min.

## 3. Results and Discussion

### 3.1. The Effect of Different Rotating Speeds of Packing and Calcination Temperature on Nano-TiO_2_

Nano-TiO_2_ synthesized at different rotating speeds of packing in IS-RPB and calcined at 500 °C is shown in Figure 3; the nano-TiO_2_ is spherical and granular. With the increase of the rotating speed of packing, the particle size decreases gradually. This is mainly because, at a higher rotating speed, the liquid microelement cut by the packing is smaller, and the liquid microelement has a faster coagulation and dispersion rate, resulting in a better microscopic mixing effect. The precipitation reaction occurs in a more uniform and higher supersaturation environment. When the rotating speed of packing is 1200 rpm, because the surface energy of small particles is higher, the dispersion and uniformity of particle sizes are obviously worse than TiO_2_ synthesized at rotating speeds of 400 and 800 rpm. In fact, the particle sizes at 400 rpm and 800 rpm are, respectively, about 25 ± 5 nm and 13 ± 3 nm from the SEM images, and the spherical granular structure is also obvious. However, due to the serious agglomeration phenomenon, there is no obvious spherical granular structure at 1200 rpm from the SEM images, and the particles have a wide particle size distribution. The better dispersion and uniform particle sizes are better for the production and application of nanomaterials, and so the best speed in the experiment is 800 rpm. TEM shows that the particle size of nano-TiO_2_ obtained under 800 rpm was about 12.5 nm.

The XRD spectra of nano-TiO_2_ synthesized at different temperatures are shown in Figure 4. In the XRD spectra, the characteristic peak of the anatase phase appears in TiO_2_ after calcination at 400 °C, but the peak pattern is weak, indicating that the crystal is not complete and only partially transformed into the anatase phase. TiO_2_ annealed at 500 °C shows a complete anatase diffraction peak, and the main peaks appear at the angles of 2θ at 25.18°, 37.90°, 48.09°, 53.96°, 62.71°, 69.12°, 70.30° and 75.00°, respectively, corresponding to the crystal planes of the anatase phase being (101), (004), (200), (105), (204), (220), (215), respectively. The results show that the amorphous TiO_2_ is completely transformed from amorphous to anatase (JCPDS No. 021/1272) [34,35]. After being calcinated at 700 °C and 800 °C, the rutile diffraction peak (2θ at 27.34° and other positions) appears. The peak pattern becomes particularly sharp after calcination at 600 °C, 700 °C and 800 °C, reflecting the increase of grain size after calcination at higher temperatures. The grain size is calculated by the Scherrer formula.
(2)$$D=\frac{k\lambda }{\beta \cos\theta }$$
where D is the grain size, k is the Scherrer constant, λ is the X-ray wavelength (0.15406 nm), β is the full width at half maximum (FWHM), and θ is the Bragg diffraction angle. The grain size of nano-TiO_2_ was calculated by using the strongest peak (2θ = 25°) [36] with its FWHM in Figure 4. As shown in Table 1, the grain sizes of TiO_2_ after calcination at 400 °C and 500 °C are 8.4 and 9.6 nm. When the temperature continues to increase, the grain size begins to increase greatly. This phenomenon is consistent with that in the studies of Rajesh et al. [37] and Phoon et al. [38], who reported that the grain size of TiO_2_ increased with the increase of calcination temperature from 523 K to 1023 K. Generally speaking, anatase TiO_2_ has good photocatalytic activity, while rutile has higher stability. In this paper, TiO_2_ calcined at 500 °C has a complete anatase phase and small grain size. Figure 3g shows the TEM image of nano-TiO_2_ after calcination at 500 °C, and the particle size is about 12.5 nm. Finally, according to the method of Surolia et al. [39], the percentage of the two phases in the nano-TiO_2_ was determined by the intensity of the peak at 2θ = 25.18° and 27.34°. The proportion of anatase (A (%)) was calculated by Formula (3):(3)$$A=\frac{100}{1+1.265\frac{I_{R}}{I_{A}}}$$
where I_A_ and I_R_ are the intensity of the anatase peak at 2θ = 25.18° and the rutile peak at 2θ = 27.34°. Crystallinity was calculated by measuring the total peak area under four major peaks at 2θ = 25.18°, 37.90°, 48.09°, 53.96° with reference to P25 as 100%. The result is shown in Table 1.

### 3.2. Characterization of the Pure TiO_2_ and Fe-TiO_2_

In order to accelerate the migration rate of electrons in TiO_2_, inhibit the recombination of the electron and hole, improve the photoresponse range of TiO_2_ and improve the photocatalytic efficiency of TiO_2_, TiO_2_ was modified by the in-situ doping of Fe^3+^ ions by precipitation at 800 rpm in IS-RPB. The catalyst was characterized after being calcinated at 500 °C.

#### 3.2.1. XRD Characterization

The crystal structures of pure TiO_2_ and Fe-TiO_2_ were characterized by XRD. As shown in Figure 5a, the positions of the diffraction peaks of pure TiO_2_ and Fe-TiO_2_ are the same, and each diffraction peak is the diffraction peak of the anatase phase (the corresponding crystal plane is marked in the diagram). Additionally, the diffraction peaks of pure TiO_2_ and Fe-TiO_2_ are the same. There are no diffraction peaks of the rutile phase and iron-related crystal phase in the XRD pattern, which indicates that the doping of Fe^3+^ does not change the crystal structure of TiO_2_, and Fe^3+^ does not form a phase alone. However, compared with the pure TiO_2_ and Fe-TiO_2_ XRD spectra, the diffraction peak intensity decreases after doping the iron, indicating that the existence of Fe^3+^ has an inhibitory effect on the crystallization of TiO_2_ [36].

#### 3.2.2. Raman Characterization

The Raman spectra of the prepared photocatalysts are shown in Figure 5b. The characteristic peaks at 141.6 cm^−1^, 396.3 cm^−1^, 516.4 cm^−1^ and 638.9 cm^−1^ correspond to the anatase phase [40,41], but there are no characteristic peaks of the rutile phase and iron-related phase except for the four peaks of the anatase phase. Compared with pure TiO_2_ and Fe-TiO_2_, it is found that the Raman shift does not change after Fe^3+^ doping, but the peak intensity and peak width change in varying degrees; in particular, the peak intensity of 5%Fe-TiO_2_ decreases and the peak width increases, indicating that Fe^3+^ replacing Ti^4+^ in the lattice results in the increase of lattice distortion and peak width, while the inhibition of crystal transformation weakens the peak intensity. This conclusion is consistent with the analysis of XRD patterns.

#### 3.2.3. XPS Analysis

XRD and Raman analysis did not find the specific existence state of iron, but both showed evidence of the existence of iron. Therefore, the synthesized Fe-TiO_2_ was analyzed by XPS to determine the specific existence of iron. Figure 6 shows the full XPS spectrum of 5%Fe-TiO_2_ and the high-resolution spectra of Ti2p, O1s and Fe2p. All electron binding energies were calibrated according to C1s (284.6 eV) of contaminated carbon. The full spectrum of 5.0%Fe-TiO_2_ is shown in Figure 6a; the main elements are Ti and O, and there are relatively few Fe elements in the sample, meaning that the detected signal of the Fe element is relatively weak. The peaks at 458.7 eV and 464.5 eV correspond to the Ti2p characteristic peaks of Ti2p3/2 and Ti2p1/2 in Figure 6b. The peaks at 529.9 eV and 531.5 eV correspond to the characteristic peaks of O1s in Figure 6c. Figure 6d shows less obvious Fe2p characteristic peaks than Figure 6a, which are the main characteristic peaks of Fe2p3/2 and Fe2p1/2 corresponding to 711.5 eV and 724.6 eV and the satellite peaks of 718.9 eV. This is consistent with the detection of Fe^3+^ by Yamashita et al. [42], which proves that iron exists in the positive trivalent form. No single iron phase is found in Raman spectra and XRD, leading us to speculate that iron could be doped into the lattice.

#### 3.2.4. TEM, HRTEM and EDX Mapping

TEM and HRTEM images of 5%Fe-TiO_2_ nanoparticles were measured to study their morphology and microstructure (Figure 7a–c). From Figure 7a,b, it can be seen that the 5% Fe-TiO_2_ particles remain the same spherical particles as the prepared pure TiO_2_, with a particle size of 11.3 ± 1.9 nm. The lattice stripes of 5%Fe-TiO_2_ particles are shown in Figure 7c, and the lattice stripe spacing of 0.352 to 0.360 nm corresponds to the (101) crystal plane of the TiO_2_ anatase phase. EDX mapping characterization was carried out to determine the doping status of Fe^3+^ in TiO_2_. From Figure 7d, it can be seen that Fe, Ti and O elements are uniformly distributed in the whole visual field, indicating that Fe has formed uniform doping in TiO_2_ and also proving that the uniform doping of Fe elements could be realized in IS-RPB.

#### 3.2.5. Photoresponse of Nano-TiO_2_

The photoresponse of prepared samples was analyzed by UV-Vis DRS and compared with commercial P25. The results are shown in Figure 8a. The curves of all the samples have a similar shape, indicating that iron did not significantly change TiO_2_. The difference of the curves lies in the difference of the downward trend and amplitude at 350 to 600 nm. The intersection of the tangent with transverse coordinates at the most obvious downward trend of the curve represents the limit of light absorption. All samples had strong light absorption below 400 nm, but P25 and pure TiO_2_ had almost no absorption at 400 to 800 nm. On the contrary, Fe-TiO_2_ shows different degrees of absorption to visible light, and the maximum optical absorption wavelength increases with the increase of Fe^3+^ content. This is probably due to the fact that Fe^3+^ precipitates uniformly in TiO_2_ during precipitation synthesis, and after calcination at high temperature, Fe^3+^ enters into the TiO_2_ lattice and uniformly replaces part of Ti^4+^, resulting in the absorption of visible light. The calculation of the band gap energy of samples by the Kubelka–Munk formula is as follows:(4)A=−lg(R)
(5)$${F(R)=(1-R)}^{2}/2R$$
(6)E=1240/λ
where A is the absorbance, R is the reflectivity, E is the band gap energy and λ is the optical wavelength. The band gap energy diagram of the sample is obtained by using (F(R)*E)^1/2^ as the longitudinal coordinate and E as the transverse coordinate, as shown in Figure 8b. The linear part of the spectrum is extended and the intersection point with the transverse coordinate is the band gap energy of the sample. The band gap energies of P25 and pure TiO_2_ are 3.12 eV and 3.06 eV, respectively. With the increase of Fe^3+^ content from 0.5% to 5.0%, the band gap energy decreases from 2.86 eV to 2.26 eV. The main reason for this is that the iron doping into the TiO_2_ lattice produces a new impurity energy level, which changes the electronic structure, leading to the decrease of the band gap energy and the absorption of visible light [43,44].

#### 3.2.6. BET Analysis

In order to analyze the surface area, total pore volume and average pore diameter of photocatalysts, N_2_ adsorption–desorption isotherms were measured, and the corresponding data are shown in Table 2. In Figure 9, the adsorption trend and desorption trend of pure TiO_2_, 1.0%Fe-TiO_2_ and 5.0%Fe-TiO_2_ prepared by IS-RPB show sharp changes at P/P_0_ between 0.6 and 0.9, indicating that isotherms are the IV-type and that the pore is mesoporous. The hysteretic loop formed by the curve is obviously H2-type, indicating that the catalyst has an ink-bottle-type pore structure [45]. P25 corresponds to the non-porous II-type isotherm. In the pore size distribution diagram and Table 2, the three catalysts prepared by IS-RPB show similar small and narrow pore size distributions and a similar total pore volume, while P25 has a wide pore size distribution and a pore size of 30.56 nm, which is the reason the specific surface area of the three catalysts is larger than P25.

### 3.3. Photocatalytic Activity

A control experiment was carried out to explore the adsorption of toluene by the catalyst and the direct decomposition of toluene by high-energy UV light; i.e., toluene was degraded by only the catalyst, only light, and both the catalyst and light (1.0%Fe-TiO_2_ was used as the catalyst). The result is shown in Figure 10a. Using only the catalyst, there is a slight decrease of the toluene concentration at the initial time, which was caused by the adsorption of the catalyst on toluene, and the concentration no longer decreases when the adsorption reaches equilibrium. Using only light, the toluene concentration remained weakly decreasing as the degradation progressed, because the photon energy at 254 nm was larger and a little toluene was decomposed. Using both the catalyst and light, toluene was greatly degraded. This is because TiO_2_ is excited by UV light, forming electrons and holes, and a series of reactions produce oxidative species such as ·OH, which can effectively decompose toluene.

Figure 10b compares the photocatalytic properties of the prepared pure TiO_2_ and Fe-TiO_2_ with commercial P25 as a control. The degradation rates of P25 and pure TiO_2_ were 56.2% and 55.4%, respectively, because the electron and holes of pure TiO_2_ were easily combined and rendered inefficient. The two crystal phases of P25 transferred some electrons between the two phases, thus exhibiting slightly higher efficiency than pure TiO_2_. When Fe^3+^ was doped, the efficiency was greatly improved. The degradation efficiencies of Fe-TiO_2_ with Fe^3+^ at 0.5%, 1.0%, 3.0%, and 5.0% were 89.9%, 95.7%, 90.5% and 74.1%, respectively. This is due to the beneficial effect of iron doping on the formation of TiO_2_, which leads to the enhancement of photocatalytic activity.

Figure 10c shows the concentration change curves of the degradation process of different initial toluene concentrations over time. When the initial toluene concentration is 105 ppm, the degradation efficiency is higher than the initial concentration at 210 ppm and 315 ppm in the same time. When the degradation efficiency is the same, the degradation time at a high concentration is often longer. However, the degradation amount increases simultaneously with the increase of initial concentration, which may be due to the increased likelihood of contact between the toluene and the catalyst, resulting in more degradation.

The result of UV-vis DRS has shown a photoresponse under visible light. In order to explore the degradation ability of the catalyst under visible light, an Xe lamp (300 W, 25.89 mW/cm^2^), the light of which was below 400 nm, eliminated by an ultraviolet cut-off filter, was used as a visible light source. Figure 10d shows the degradation ability of P25, pure TiO_2_ and two Fe-TiO_2_. P25 and pure TiO_2_ showed no degradation ability under visible light; the minimal decrease of toluene was caused by adsorption. On the contrary, both 1.0%Fe-TiO_2_ and 5.0%Fe-TiO_2_ showed degradation ability under visible light. However, the photocatalytic efficiency decreased from 95.7% and 74.1% under UV light to 54.9% and 47.7%, respectively. The possible reason for this is that the addition of iron leads to the absorption of visible light by TiO_2_. However, because the energy of visible light is lower than that of UV light, only part of the light was used, and so the photodegradation efficiency is lower than that under UV light.

### 3.4. Possible Mechanism of Photocatalytic Enhancement

The degradation experiment of toluene showed that the degradation efficiency of Fe-TiO_2_ was higher than that of TiO_2_ and P25. To explain this phenomenon, photoluminescence spectra (PL) were employed under the excitation of light of the same wavelength (254 nm) as the degradation experiment (Figure 11). The main peak positions of all catalysts were the same, distributed at 400, 414, 442, 455 and 471 nm, and no new emission peaks appeared at the PL of Fe-TiO_2_. The peak at 400 nm corresponds to the recombination of electrons and holes of TiO_2_; the other peaks were probably produced by oxygen vacancies or band edge free excitons [46,47]. The peak intensity of Fe-TiO_2_ at 400 nm is obviously weakened, which shows that the doping of iron has an obvious inhibitory effect on the recombination of electrons and holes. The weakening trend of peak intensity is completely consistent with the degradation at different catalysts, which shows that the doping of Fe^3+^ has an optimum concentration and that 1% Fe may be the optimal doping concentration. The doping level was insufficient at 0.5%Fe-TiO_2_, and Fe^3+^ may become the recombination center of electrons and holes, leading to reduced catalytic performance, because of the high iron content at 5%Fe-TiO_2_.

Figure 12 shows a possible mechanism of photocatalytic enhancement. It has been shown that the iron in Fe-TiO_2_ exists in the form of Fe^3+^ and maintains a highly uniform distribution, while the radius of Fe^3+^ is close to that of Ti^4+^, which is conducive to the formation of uniform substitution doping into the lattice [22], resulting in a doping level with low width. The generation of this energy level leads to the red shift of light absorption. At the same time, Fe^3+^ exhibits a transformation of Fe^3+^→Fe^4+^ and Fe^3+^→Fe^2+^, which can capture and transfer photogenerated electrons and holes, respectively [22,36,46]. The recombination of them is prevented, which improves the quantum efficiency, which is consistent with the results shown by PL spectra.

## 4. Conclusions

In summary, a high-gravity precipitation method for the synthesis of nano-TiO_2_ was developed in this work. Through the exploration of the synthesis process, TiO_2_ with anatase crystals, smaller particle sizes, better dispersion and average particle sizes of 12.5 nm was obtained in IS-RPB. When the calcination temperature was 500 °C, nano-TiO_2_ had a better anatase crystal and maintained a smaller particle size. At the same time, Fe-TiO_2_ was synthesized in situ for the degradation of toluene as a VOC. The results show that the Fe-TiO_2_ anatase phase remains intact, and the variation of the Raman peak width and intensity shows evidence of the existence of iron. XPS and EDX mapping shows that iron was uniformly distributed in TiO_2_ and exists in the positive trivalent form. When the Fe^3+^ was increased from 0.5% to 5.0%, the band gap energy of the catalyst was reduced from 3.06 eV in undoped Fe to 2.86 eV and 2.26 eV. Fe^3+^ not only broadens the photoresponse range but also effectively suppresses the recombination of the electron and hole. As a result, the catalytic activity was enhanced. The degradation rate of 1.0%Fe-TiO_2_ to toluene of 105 ppm reached 95.7% after 4 h under UV light.

## Figures and Tables

**Figure 1 nanomaterials-09-01173-f001:**
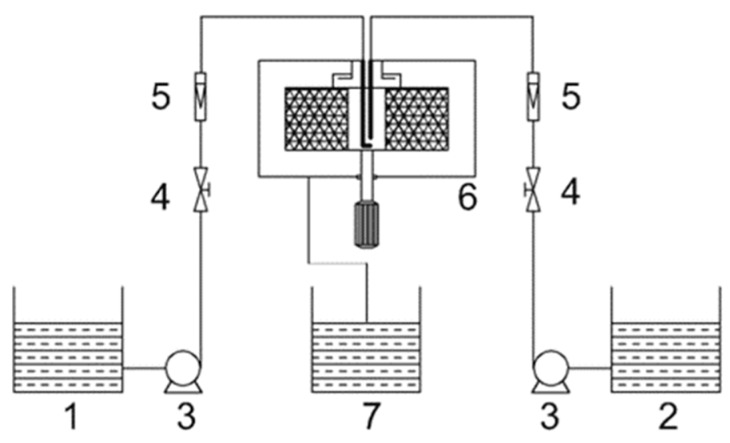
Experimental setup: 1, 2: liquid storage tank; 3: pump; 4: valve; 5: rotormeter; 6: impinging steam-rotating packed bed (IS-RPB); 7: collection tank.

**Figure 2 nanomaterials-09-01173-f002:**
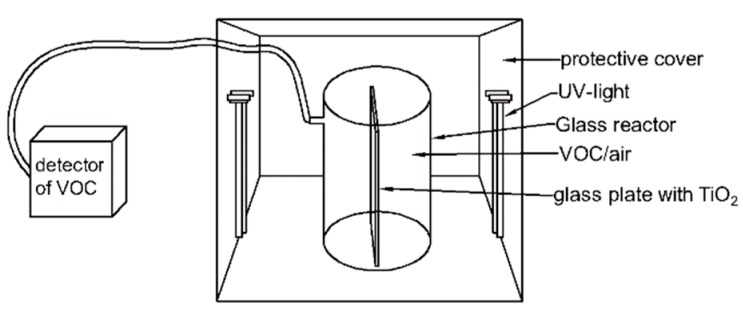
The self-made quartz reactor.

**Figure 3 nanomaterials-09-01173-f003:**
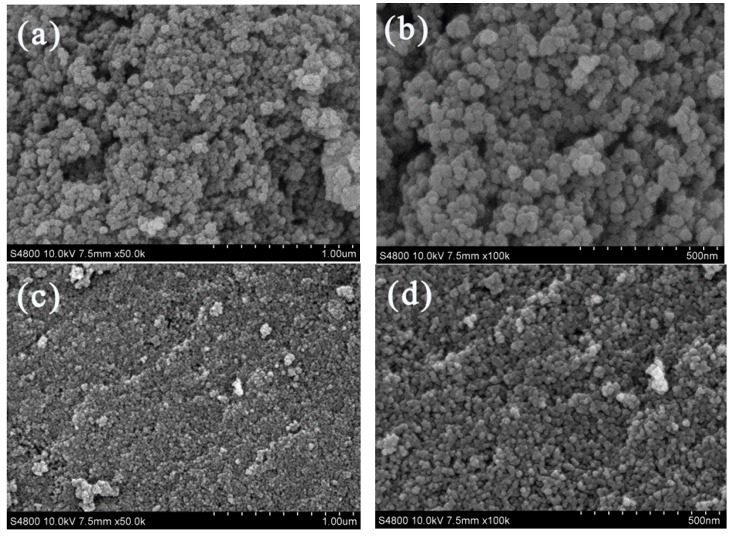
SEM images of nano-TiO_2_ at rotating speeds of the packing of 400 rpm (**a,b**), 800 rpm (**c,d**), and 1200 rpm (**e,f**). TEM images of nano-TiO_2_ at a rotating speed of the packing of 800 rpm (**g**).

**Figure 4 nanomaterials-09-01173-f004:**
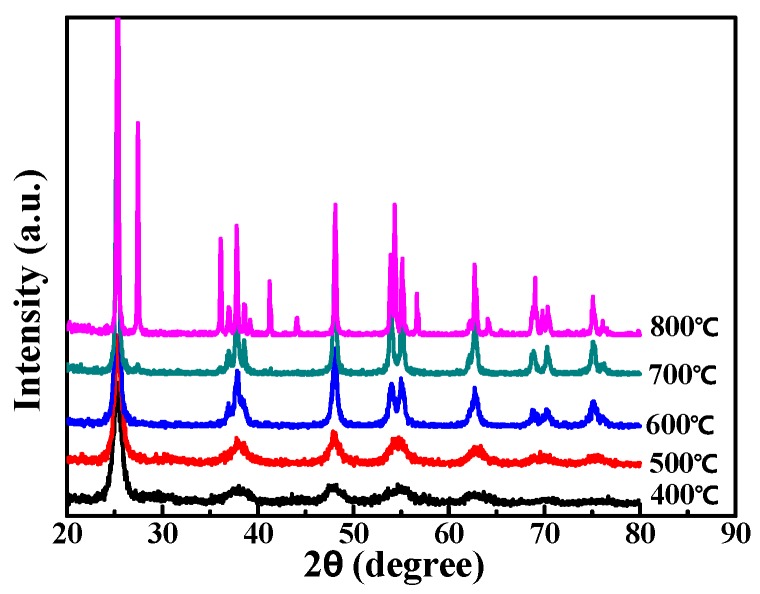
XRD spectra of nano-TiO_2_ calcined at different temperatures.

**Figure 5 nanomaterials-09-01173-f005:**
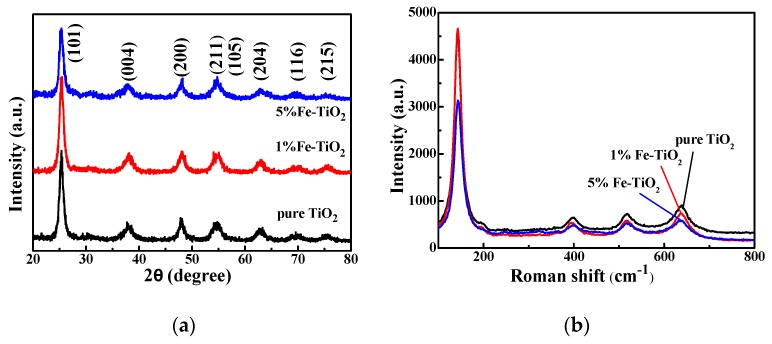
(**a**) XRD spectra of pure TiO_2_ and Fe-TiO_2_; (**b**) Raman spectra of pure TiO_2_ and Fe-TiO_2_.

**Figure 6 nanomaterials-09-01173-f006:**
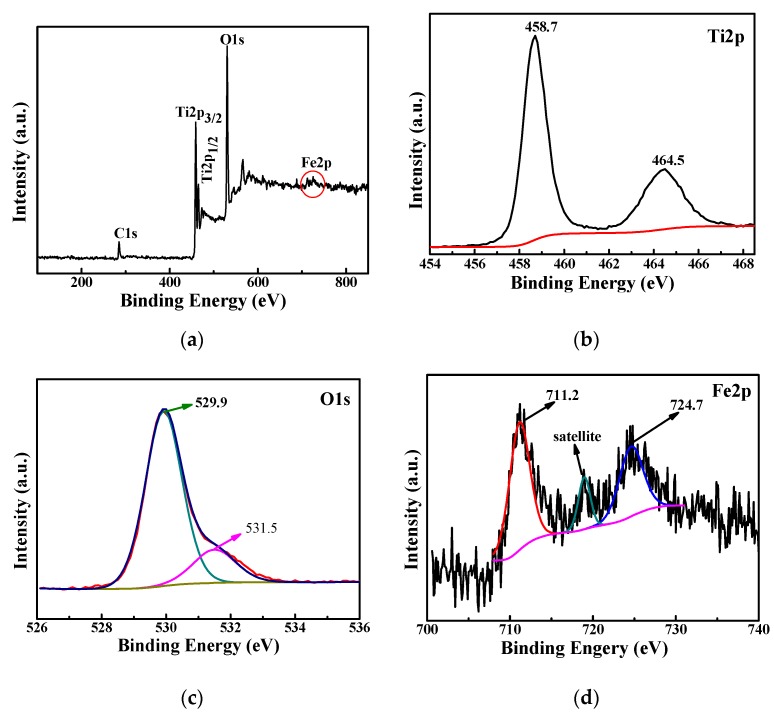
XPS spectra of 5%Fe-TiO_2_ (**a**) and the high-resolution spectra of Ti2p (**b**), O1s (**c**) and Fe2p (**d**).

**Figure 7 nanomaterials-09-01173-f007:**
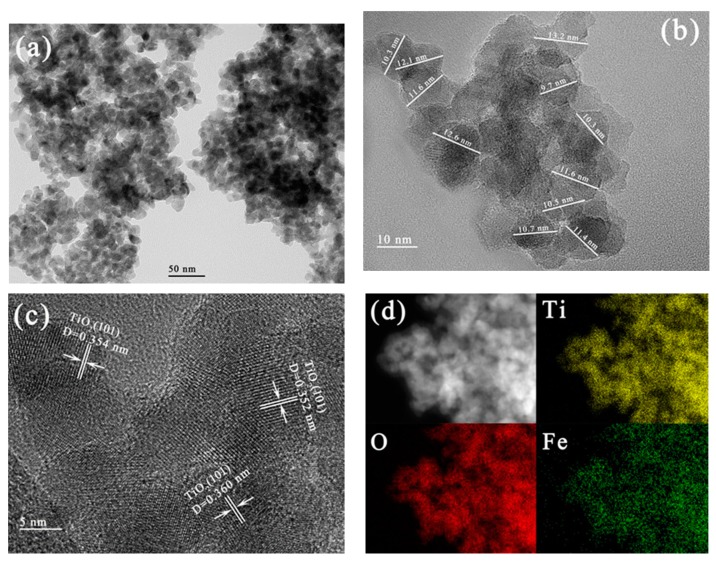
TEM (**a**) and HRTEM (**b,c**) images, energy dispersive X-ray (EDX) mapping (**d**) of 5%Fe-TiO_2_.

**Figure 8 nanomaterials-09-01173-f008:**
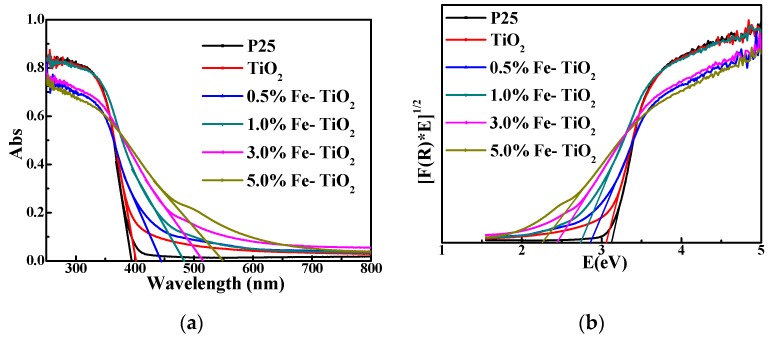
(**a**) UV-vis diffused reflectance spectroscopy (DRS) spectra and (**b**) band gap energies of P25, pure TiO_2_ and Fe-TiO_2_.

**Figure 9 nanomaterials-09-01173-f009:**
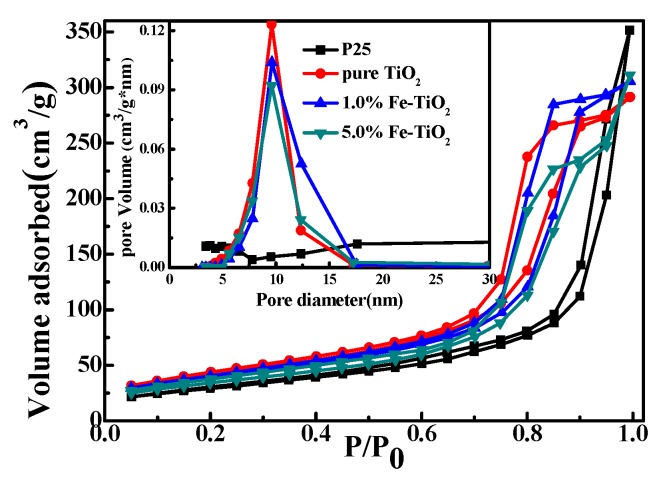
N_2_ adsorption–desorption isotherms and the pore size distributions of photocatalysts.

**Figure 10 nanomaterials-09-01173-f010:**
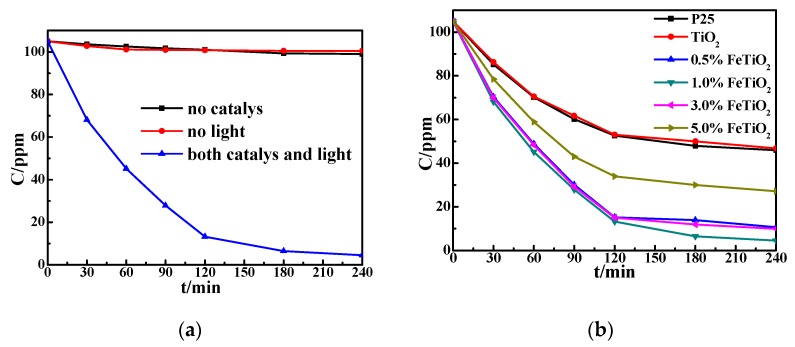
Photodegradation of toluene in the control experiment (**a**), using different catalysts (**b**), under different initial toluene concentrations (**c**) and using visible light (**d**).

**Figure 11 nanomaterials-09-01173-f011:**
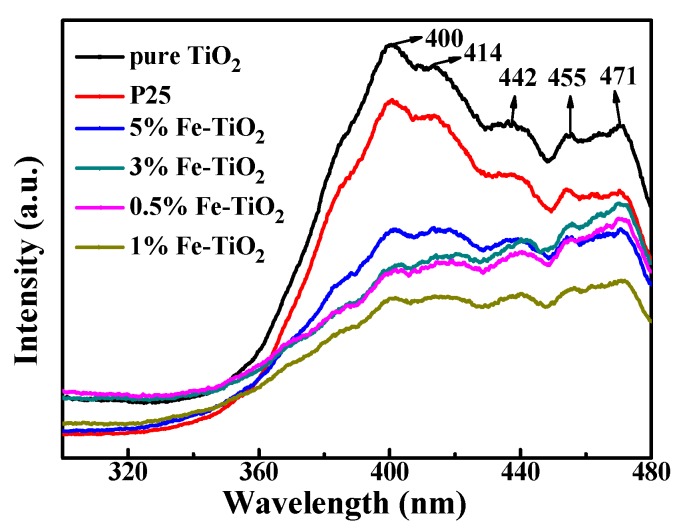
Photoluminescence (PL) spectra of different catalysts excited by UV light of 254 nm.

**Figure 12 nanomaterials-09-01173-f012:**
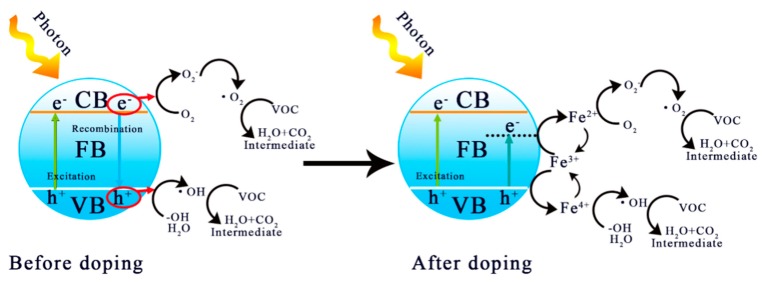
Possible mechanism of photocatalytic enhancement of Fe^3+^-doping.

**Table 1 nanomaterials-09-01173-t001:** Size of nano-TiO_2_ and crystallinity calculated by XRD (X-ray diffraction).

Temperature (°C)	FWHM (°)	β’ (rad)	XRD Size (nm)	Anatase (%)	Rutile (%)	Crystallinity (%)
P25	-	-	-	77%	23%	100%
400	0.951	0.0166	8.4	100%	0%	70%
500	0.837	0.0143	9.6	100%	0%	82%
600	0.488	0.0085	16.5	100%	0%	110%
700	0.308	0.0054	26.0	95%	5%	97%
800	0.163	0.0028	50.2	67%	33%	81%

**Table 2 nanomaterials-09-01173-t002:** Surface area, total pore volume and average pore diameter of photocatalysts.

Photocatalysts	Surface Area (m^2^/g)	Total Pore Volume (cm^3^/g)	Average Pore Diameter (nm)
P25	105.96	0.5438	30.56
TiO_2_	148.47	0.4508	9.58
1.0%Fe-TiO_2_	140.18	0.4643	9.60
5.0%Fe-TiO_2_	123.50	0.4813	9.58

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
