# Peer review of "Preparation of TiO2 and Fe-TiO2 with an Impinging Stream-Rotating Packed Bed by the Precipitation Method for the Photodegradation of Gaseous Toluene"

_nanomaterials, 2019, doi:10.3390/nano9081173_

Round 1

Reviewer 1 Report

 Preparation of TiO2 and Fe-TiO2 through Impinging  Stream-Rotating Packed Bed by Precipitation Method  for Photodegradation of Gaseous Toluene

Present manuscript discusses the synthesis of nano-TiO2 by high gravity precipitation method and doped with Fe to enchance the photocatalytic activity of the photocatalysts by absorbing the visible light. I would like to consider the manuscript for revision at the present level. Based on the revision manuscript can be further considered for the publication.

Authors are requested to consider following point during revision of this manuscript.

1.    Authors need to discuss write the particular techniques used for characterization of catalyst in abstract, I would like to suggest the authors to remove “so on” from the abstract.

2.    As in intro page 2, line 78-80, author mentioned the doping of Fe 3+ may enhance the response of TiO2 to visible light then why authors have provided the photocatalytic results of the degradation in UV light, it is very good if they can used visible light for degradation and it is not harmful.

3.    Page 2, line 67 : the word “mental” need to be corrected

4.    Page 3, line line 99, correct word “diameterin”

5.    Page3, line 113, give space after no

6.    Page 4, line 147 correct spelling of toluene

7.    Please correct the subsection titles 2.2. Synthesis of nano-TiO2 and Fe-TiO2 and 2.3. Synthesis of Fe-TiO2 nanoparticles

8.    Synthesis of nano-TiO2 is already explained in section 2.2 then why the section 3.1 is titled as synthesis ofnano-TiO2, it will better if they given subtitle as “characterization of nano-TiO2”

9.    Author need to add the percentage crystallinity of anatase peak of the synthesised nano-TiO2 at different temperature. Author can see and cite the ref. to calculate the percentage of crystallinity like Ind. Eng. Chem. Res.2007,46, 19, 6196-6203.

10. In discussion of crystallite size with respect to temperature, author need to add some supportive ref such  as Industrial & Engineering Chemistry Research 2006, 45 (3), 922-927

11. Section 3.2 need to be subtiled as “ Characterization of Fe-TiO2

12. Section 3.2.2 “Roman” need to be replaced by “Raman”

13. As the catalysts have the visible light capability why UV light is used to conduct the photocatalytic experiment, It will better if author can provide both the experimental result of photocatalytic degradation experiment carried under uv and visible light

14. Caption given in Figure 11 are not correct, needs correction

15. Author can add some more ref on Toluene degradation such as Journal of the Air & Waste Management Association 2015, 65 (3), 365-373 and some other work carried out by W. K. Jo

16. Authors need to add the surface area of the synthesised photocatalytic materials.

Reviewer 2 Report

1. The manuscript should be carefully edited. For example, XRD, and XPS should be spelled out in the first place in the abstract and context.

2. The Abstract should provide some important data.  

3. p.10,  Fig. 8 (a), it seems no big difference among the six samples. It need more discussion and explanation would have been interesting see.  

4. The Synthesis for the preparation of nano-TiO2 and Fe-TiO2 should be briefly described in the manuscript with the calcination temperature.

4. The English usage should be improved.

Reviewer 3 Report

Zeng et al. Describe their finding on the Preparation of TiO2 and Fe-TiO2 through Impinging Stream-Rotating Packed Bed by Precipitation Method for Photodegradation of Gaseous Toluene. The manuscript is suitable for publication in Nanomaterials after the author address the following minor revisions:

Light intensity (mW/cm2) should be given in the experimental part instead of the giving power of the UV lamp. Because the distance between the sample and UV source can affect the photocatalytic measurement.

Figure 3e and 3f are not clear. Authors should provide clear SEM images.

Authors claim “The better dispersion and uniform particle size are more powerful for the production and application of nanomaterials, so the best speed in the experiment is 800 rpm. TEM showed that the particle size of nano-TiO2 obtained under 800 rpm was about 12.5 nm” In order to get anatase minimum 500℃ heat-treatment is needed. Authors should clarify that How average particle sizes change after heat-treatment and there is any agglomeration or not after heat-treatment.

Particles size distribution should be given to understand how average particle sizes change according to the rotating speed of the packing. SEM images are not clear to understand particle size and possible agglomeration.

Raman spectra should be given in section 3.2.2.

Figure 11 should be checked. The labels of the schematic are wrong. “After” and “Before” should exchange.

Reviewer 4 Report

The authors have discussed the effect of doping on the photoactivity of the titania in in which the iron plays the role of electron mediator. Therefore, it has the merit of publication in the hournal. However, a minor revision is needed:

1- SEM images are not well-arranged.

2-why Fe-titania is not studied at different temperatures?

3-English needs to be revised.

4-Why SEM of Fe-TiO2 with different temperatures are not performed?

Round 2

Reviewer 1 Report

Authors have tried to revise the manuscript as per the reviers suggestion. I would like to recommend the publication of this manuscript, after correction of reference 39 authors name (Surolia P. K). 
